# Polymeric Nanoparticles Decorated with Monoclonal Antibodies: A New Immobilization Strategy for Increasing Lipase Activity

**Laura Chronopoulou** [1], **Viviana Couto Sayalero** [1], **Hassan Rahimi** [2], **Aurelia Rughetti** [2] **and Cleofe Palocci** [1,*]

1 Department of Chemistry, University of Rome La Sapienza, Piazzale A. Moro 5, 00185 Rome, Italy; laura.chronopoulou@uniroma1.it (L.C.); coutosayalero.1655568@studenti.uniroma1.it (V.C.S.)

2 Department of Experimental Medicine, University of Rome La Sapienza, Viale Regina Elena 324, 00161 Rome, Italy; hassan.rahimi@uniroma1.it (H.R.); aurelia.rughetti@uniroma1.it (A.R.)

* Correspondence: cleofe.palocci@uniroma1.it; Tel.: +39-06-49913317

**Abstract:** Recent advances in nanotechnology techniques enable the production of polymeric nanoparticles with specific morphologies and dimensions and, by tailoring their surfaces, one can manipulate their characteristics to suit specific applications. In this work we report an innovative approach for the immobilization of a commercial lipase from *Candida rugosa* (CRL) which employs nanostructured polymeric carriers conjugated with anti-lipase monoclonal antibodies (MoAbs). MoAbs were chemically conjugated on the surface of polymeric nanoparticles and used to selectively adsorb CRL molecules. Hydrolytic enzymatic assays evidenced that such immobilization technique afforded a significant enhancement of enzymatic activity in comparison to the free enzyme.

**Keywords:** lipase; monoclonal antibodies; enzyme immobilization; polymeric nanoparticles

## 1. Introduction

Enzymes have been extensively employed in many applications, becoming favored catalysts for green chemistry approaches thanks to their high substrate specificity and mild reaction conditions [1,2]. Microbial lipases constitute an important group of biotechnologically valuable enzymes, mainly because of their versatility and the ease of large-scale production. They are widely diversified in their enzymatic properties and substrate specificity, which make them very attractive for industrial applications. For this reason lipases are often used in organic synthesis applications, being able to ensure high stereoselectivity and regioselectivity [3,4]. *Candida rugosa* lipase (CRL) is the most commonly used catalyst in the bioindustry since it is produced by a GRAS mold [5,6]. The main drawback in employing lipases is their lack of stability during long-term reactions or storage. On this basis, immobilization on a solid support is considered a major achievement for industrial applications because the enzyme can retain its activity and can be easily recovered. Recently, the possibility of immobilizing or conjugating enzymes onto nanoparticles has been described, alongside the advantages of high enzyme loading, improved stability and activity [7,8]. Previous studies carried out within our research group highlighted how CRL immobilization on polymeric nanoparticles (NPs) allows a stabilization of its active conformation with respect to the inactive one, with a consequent increase of its activity [9]. The use of nanostructured carriers for the immobilization of enzymes would also make it possible to increase the surface available for binding with the enzymatic proteins and to stabilize the three-dimensional conformation of the protein itself, particularly when using enzymes in organic medium.

Protein attachment to a solid support with a preferable orientation can effectively avoid its denaturation and keep its active sites fully exposed to solution, thus maximally preserving its bioaffinity or bioactivity. In particular, one interesting methodology of

protein immobilization has been proposed by Minarik et al. [10] using enzyme fusion as the binding site. In this case, a protein of interest is fused to an enzyme that reacts selectively with an immobilized substrate analogue or inhibitor to form a covalent bond. Following this approach, enzyme-mediated site-directed covalent immobilization techniques have been developed for immunosensing applications [11–14]

As is well known, the use of monoclonal antibodies (MoAbs) acquired relevance in basic immunological research in the past, finding various applications ranging from industrial technologies in the biotechnological field to diagnostics and therapy [15,16]. In the industrial field, MoAbs are used as a purification tool for proteins of biotechnological interest (i.e., cytokines, growth factors). The activity of an immobilized antibody, i.e., its target binding ability, finds its origin in its Fab domain, which is (i) accessible, i.e., has an outward orientation from the interface; and (ii) biologically active, i.e., has a molecular conformation with a low dissociation constant (Kd) for the target molecule. The target binding activity of antibodies immobilized at a solid–liquid interface has been studied as a function of immobilization strategy, molecular orientation, surface crowding of the antibodies and solid surface properties [17–20]. On this basis, the possibility of chemically functionalizing enzyme supports by using monoclonal antibodies as specific receptors seems to be a novel approach for specific enzyme binding.

In this work, we used, for the first time, anti-lipase MoAbs, chemically conjugated on the surface of polymeric NPs as specific lipase binding receptors, to selectively immobilize CRL through non-covalent bonds starting from commercial enzyme preparations. The catalytic activity of enzyme/antibody complexes in solution has already been demonstrated by our group [21]. In particular, poly-(D,L-lactic-co-glycolic) acid (PDLLA) NPs, synthesized by using a patented methodology, were prepared. PDLLA was chosen since it is a widely used biopolymer in many biotechnological applications, its production is more sustainable than many synthetic polymers and it is commercially available. Moreover, PDLLA has carbonyl groups in its backbone, which can be easily used in order to functionalize it. Anti-lipase MoAbs were covalently coupled via EDC/NHS chemistry on the surface of PDLLA NPs, with a strategy chosen because of its extensive use in biofunctionalization due to its robustness and minimal need for chemical modification [22,23]. This immobilization approach based on a double-step procedure stabilizes enzyme molecules and avoids the reduction of enzyme activity generally achieved after a direct covalent binding on the immobilization support.

A dimensional characterization of native PDLLA NPs and conjugated with the antibodies was carried out through dynamic light scattering (DLS) and scanning electron microscopy (SEM). Finally, the lipolytic activity of CRL, free or immobilized on PDLLA NPs conjugated with MoAbs, was evaluated.

## 2. Results and Discussion

### 2.1. MoAb Purification Procedure

First of all, we approached the purification of anti-CRL MoAbs from ascitic fluid produced by Balb/C mice inoculated peritoneally with hybridoma cells producing MoAbs [21]. The purification procedure was carried out by affinity chromatography separation, using G protein. The different elution fractions present different MoAb concentrations, as reported in Table 1. We started the separation from an initial volume of 3 mL of ascitic fluid, from which we recovered 4 mg of purified MoAbs, according to Bradford assay results. The evaluation of total protein content in the ascitic fluid and MoAb-rich fraction are reported in Table 2. The high purification factor obtained (38) demonstrates well the effective MoAb purification from ascitic fluid. The MoAb employed was shown to selectively recognize CRLs, but not lipase from other yeasts [21].

**Table 1.** Characteristics of purified MoAb fractions.

| Sample | MoAb Concentration (µg/mL) | Volume (mL) | MoAb Mass (mg) |
|--------|---------------------------|-------------|----------------|
| 1 | $3.07 \pm 0.01$ | 2.00 | $6.14 \pm 0.01$ |
| 2 | $400.00 \pm 0.05$ | 3.50 | $1400.00 \pm 0.05$ |
| 3 | $1282.20 \pm 0.05$ | 2.00 | $2564.00 \pm 0.05$ |

**Table 2.** Protein content of the ascite and final purification factor of MoAbs.

| | |
|---|---|
| Ascite protein content | 20 mg/mL |
| MoAb total concentration | 0.529 mg/mL |
| Purification factor | 38 |

### 2.2. Dimensional and Morphological Characterization of PDLLA NPs

The dimensional and morphological characterizations of PDLLA NPs were performed with DLS and SEM measurements, respectively. According to DLS data, the hydrodynamic diameter of PDLLA NPs was 270 nm (with a polydispersion index of 0.125), as shown in Figure 1. As expected, the conjugation of MoAbs on PDLLA NPs surface did not significantly influence NP dimensions (data not shown). SEM micrographs of PDLLA NPs are shown in Figure 2. The NPs are spherical, with sizes comparable to those obtained by DLS. The zeta potential of PDLLA NPs was $-35 \pm 2$ mV, and it was not significantly modified by MoAb conjugation ($-38 \pm 2$ mV). This was probably due to the low MoAb conjugation on the surfaces of PDLLA NPs.

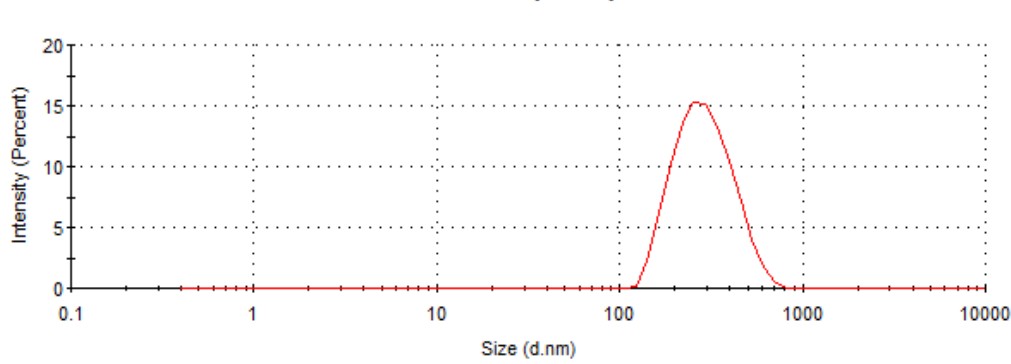

**Figure 1.** Size distribution by intensity of PDLLA NPs obtained by DLS measurements.

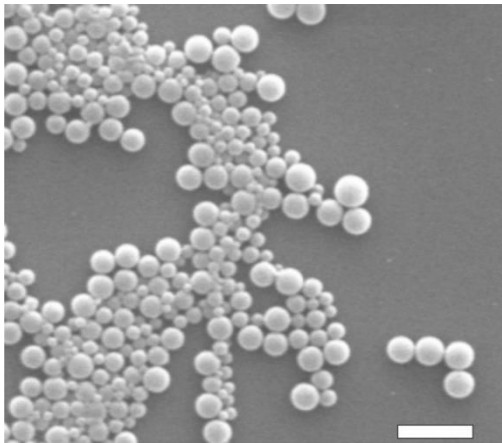

**Figure 2.** SEM micrograph of PDLLA NPs. Scale bar: 1 µm.

### 2.3. MoAb Conjugation to PDLLA NPs

For PDLLA NP functionalization, we initially used a MoAb solution with a concentration of 4 μg/μL and employed the chemical procedure of EDC/NHS functionalization reported in the experimental section. On the basis of the amount of unconjugated MoAb recovered in the supernatant, we calculated that up to 80% of the initial MoAb was bound to the surface of PDLLA NPs, with a weight ratio of 41.4 μg MoAb/mg PDLLA. No investigation on MoAb orientation (from Fc or Fab side chains) on the surfaces of PDLLA NPs was carried out at this stage, although such investigations could be useful in order to establish the "active" amount of MoAbs exposed with the correct orientation to efficiently interact with lipase molecules.

### 2.4. Lipase Adsorption on PDLLA@MoAb NPs and Lipolytic Activity Evaluation

The functionalized PDLLA NPs were used as carriers for lipase adsorption by using the MoAbs conjugated on NPs as "active ends" able to complex, by specific Ab/Ag interactions, lipase molecules. With this aim, a fixed amount of PDLLA@MoAb NPs (1.5 mg) was employed as a support for CRL immobilization. We decided to use an initial lipase concentration of 186 μg/μL with a MoAb/lipase weight ratio equal to 1/3. Lipase adsorption on the surfaces of PDLLA@MoAb NPs could occur according to different interactions: a specific antibody/antigen interaction or a non-specific one through non-covalent interactions with the surface of PDLLA NPs. The evaluation of the total amount of adsorbed lipase was achieved by testing the protein content in solution after enzyme adsorption. The amount of enzyme bound to PDLLA@MoAb NPs was 10% of the initially used amount (5 μg out of 50 μg). The very low enzyme quantity could have been the result, as expected, of a not totally specific bond of lipase molecules on the PDLLA NPs surfaces. These lipase molecules bound aspecifically could be reduced by increasing the antibody immobilization level on the surfaces of PDLLA NPs.

As reported in our previous work [21], the CRL–MoAb complex is active on synthetic esters as well as on natural substrates (triglycerides). In this work, we chose to test the activity of free or adsorbed CRL on PDLLA NPs or PDLLA@MoAb NPs by using tributyrin as a model substrate, in order to be able to compare our results with those we previously reported. As shown in Table 3, and as also previously reported by the authors [9], CRL immobilized on PDLLA NPs significantly increased its activity in comparison with the free CRL during hydrolysis reactions in standard conditions. It was proposed, on the basis of FTIR spectra, that such behavior could result from lipase conformational stabilization according to a 3D structure rearrangement upon adsorption on polymeric NPs. When CRL was complexed with MoAbs, the results indicated that the binding of the immunoglobulins to lipase molecules did not affect the lipolysis in solution; furthermore, lipases were activated during catalysis, as we have previously reported [21]. When we tested the activity of CRL immobilized on the surface of PDLLA@MoAb NPs, we found that it was 8.75 times higher than that obtained with the free enzyme. Notably, despite the relatively small amount of immobilized CRL, PDLLA@MoAb NPs were able to induce a significant increase in lipolytic activity.

**Table 3.** Lipolytic activities of free or immobilized CRL.

| Sample | Enzymatic Activity UI/mg |
|---|---|
| Free CRL | 0.216 ± 0.010 |
| CRL complexed with MoAb | 0.498 ± 0.009 |
| CRL immobilized on PDLLA NPs | 0.523 ± 0.012 |
| CRL immobilized on PDLLA@MoAb NPs | 1.898 ± 0.018 |

It is interesting to note that the epitope recognized by the employed MoAb involved the N-glycosylated Asn341 amino acid residue [21] that appears to contribute to the lipase lid opening [24].

In our previous work, we demonstrated that CRL can be activated by using different strategies. Its bond to MoAbs was shown to increase enzyme activity (~2.3 fold). In this case, the bond is specific, involving specific amino acids in the variable portion of the MoAb light chain [21]. In a later study, we demonstrated that polymeric nanoparticles can induce an activation of CRL through a shift in the conformational equilibrium of the enzyme towards its active form (~2.4 fold) [9]. In the latter case, the bond between NPs and CRL was not specific. Though based on different types of interactions, these two strategies roughly produce the same quantitative result in activating CRL molecules.

In the present work, we developed a biohybrid system, employing PDLLA NPs for the covalent immobilization of anti-CRL MoAbs. This new system is significantly more efficient in activating CRL molecules, affording an 8.5-fold increase in lipolytic activity. The mechanistic explanation of this effect is probably complex and would deserve further dedicated investigations, but we can make a few reasonable considerations.

First of all, it is clear that the observed effect is not a simple addition of the already described effects of PDLLA NPs and MoAbs on CRL activation. On the contrary, the different components of the biohybrid system are able to exert a synergic effect that results in a strong activation of CRL molecules. Different contributions to the final observed effect may be envisaged. First of all, it has been widely reported [25] that NPs, thanks to their highly curved surfaces, are able to induce conformational changes in proteins that are adsorbed or bound on such surfaces, which may affect their biological properties and behavior. In our case, the bond of MoAbs on PDLLA NPs may positively affect MoAb recognition of CRL molecules and the way they bind to each other. Moreover, MoAb immobilization on a hydrophobic surface will likely increase the local concentration of both active molecules and lipid substrates, resulting in a significant increase in lipolytic activity.

## 3. Materials and Methods

### 3.1. Materials

*Candida rugosa* lipase (CRL, type VII, ≥700 U/mg solid), tributyrin (≥99%), phenolphthalein, NaOH, protein G agarose >98% (HPLC and SDS-PAGE), PDLLA, dimethylformamide (DMF), acetone 99.5%, ethanol standard for GC, cellulose acetate dialysis bags (Avg. flat width 25 mm (1.0 in.), MWCO 12,000 Da), glycine 99%, and all other chemicals and solvents were purchased from Sigma-Aldrich (Saint Louis, MO, USA) and used as received.

### 3.2. MoAb Preparation and Purification Procedures

Anti-lipase MoAbs were produced as previously described [21]. Briefly, splenocytes from CRL-immunized Balb/c mice were isolated and fused with murine myeloma NS-2 cells. Hybridoma cultures were screened in ELISA for antibody production against CRL used as immunogen.

For the purification of MoAbs from mouse ascites, an affinity chromatography separation was employed, using protein G immobilized on agarose as the stationary phase [26].

Protein G (100 mg) was hydrated for 30 min with 2.5 mL of deionized water to obtain gelation. The resin obtained was then packed in a column (the final volume of the packed column was 0.5 mL) and balanced with phosphate buffer (PBS 20 mM) at pH 7.0, eluting it by using a buffer quantity equal to five volumes of the column (2.5 mL). A fixed amount of unpurified MoAbs was inserted into the column up to the saturation when all the active sites of the stationary phase selectively interact with the antibodies. The sample was balanced in the column using three volumes of PBS, and the eluate was collected. MoAb isolation was obtained by lowering the column buffer pH through the addition of a 100 mM glycine solution (pH 2.7). After MoAb elution, an appropriate volume of Tris 1 M buffer (pH 9.1) was added to restore neutral pH. The process was repeated several times, both with the eluate produced by washing the column with PBS, following the insertion of the sample, as well as with new ascite samples containing MoAbs.

### 3.3. Lipolytic Assay of Free and MoAb-Conjugated Enzymes

The activity of free and MoAb-conjugated lipase preparations was determined according to a standard hydrolysis assay, using tributyrin as the substrate. Free or MoAb-conjugated CRL was dispersed in 2.5 mL of PBS (pH 7.4, 0.1 M), to which 0.5 mL of tributyrin was added. The mixture was incubated at 37 °C under magnetic stirring (600 rpm) for 30 min. Hydrolysis was stopped by adding 3.0 mL of 1:1 acetone/ethanol mixture, and the reaction mixture was titrated with 0.01 M NaOH in the presence of phenolphthalein as an indicator using an automatic titrator Mettler Toledo, Columbus, OH, USA). A sample with no enzyme was prepared in the same way and used as reference. The lipolytic activities of the samples were expressed as international units (UI) corresponding to the acid μmol produced per minute and normalized for the amount of enzyme, according to the following equation:

$$UI/mg\ CRL = (A - B)N \times 1000/(t \times m_{CRL}) \tag{1}$$

A = NaOH mL used to titrate the sample;
B = NaOH mL used to titrate the reference;
N = NaOH concentration;
T = incubation time;
$m_{CRL}$ = amount of free or MoAb-conjugated enzyme present in the sample.

### 3.4. Bradford Assay

A Bradford assay was used for the quantitative analysis of proteins in solution [27]. Bovine serine albumin (BSA) was used as the standard to prepare a calibration curve that followed the equation: $y = 0.0104x - 0.0273$ ($R^2 = 0.9921$). Samples were analyzed according to the Micro 2 mL Assay Protocol provided by the manufacturer.

### 3.5. Synthesis of PDLLA NPs

PDLLA NPs were prepared by using a method patented by our research group [28]. Briefly, 75 mg of PDLLA was dissolved in 15 mL of DMF. Then, 5 mL of the polymeric solution was placed into a dialysis bag of cellulose acetate. The membranes were then immersed in 100 mL of distilled water and incubated for 120 h at 4 °C. Then, the polymer was recovered by centrifugation (14,000 rpm, 4 °C, 15 min), washed twice with $H_2O$ and freeze-dried.

### 3.6. DLS and Zeta Potential Measurements

Dynamic light scattering (DLS) and zeta potential measurements were carried out on aqueous suspensions of PDLLA NPs, PDLLA@MoAb NPs and PDLLA@MoAb@CRL NPs, using a Nano Zetasizer (Malvern Instruments, Malvern, UK). The experimental conditions were the following: a helium–neon laser operating at 633 nm, a fixed scatter angle of 173°, constant temperature T = 25 °C. Non-negative least-squares (NNLS) or CONTIN algorithms, supplied with the instrument software, were used to fit correlation data. The average hydrodynamic radius of the diffusing objects was calculated from the diffusion coefficient D and the Stokes–Einstein relationship, $R = (K_BT)/(6\pi\eta D)$, where $K_BT$ is the thermal energy and η is the solvent viscosity. The particles' z-potential was obtained from the measured mobility μ by using the Smoluchowski relation.

### 3.7. SEM Measurements

Aqueous suspensions of NPs samples were deposited on aluminum stabs, air-dried and analyzed with a Auriga 405 microscope (Zeiss, Jena, Germany) at low extracting voltage (1.5–4 kV) and current (7.5 m aperture), as well as at a very low working distance (≈1 mm), in order to improve the quality of the signal received by the in-lens detector.

### 3.8. PDLLA NPs' Functionalization with MoAbs

For PDLLA NPs' functionalization with anti-lipase MoAbs, a procedure based on EDC/NHS chemistry was used [29]. This procedure activates the carboxylic groups of PDLLA, which can thus bind to the amine groups of the lysine residues of antibody molecules.

A total of 5 mg of PDLLA NPs was dispersed in 1 mL of MES buffer solution 0.1 M containing NaCl 0.5 M under magnetic stirring for 10 min.

For the activation of the polymer, 1 mL of EDC/NHS solution, with molar ratio 2:5, was added to the buffer solution containing the polymer and kept under stirring for 15 min. Then, 62.5 μL of MoAb solution (concentration 400 μg/mL) was subsequently added for an antibody–PDLLA weight ratio of 1:20 (0.25 mg:5 mg). After the addition of the antibody, the suspension was kept under stirring for 2 h. Then, it was centrifuged and the supernatant was analyzed according to the Bradford assay in order to quantify the amount of unconjugated antibodies. The MoAb-functionalized NPs (PDLLA@MoAb NPs) were then washed twice with deionized water, centrifuged, freeze-dried and stored at 4 °C.

### 3.9. Lipase Adsorption on PDLLA@MoAb NPs

To adsorb CRL molecules on the surface of PDLLA@MoAb NPs, 1.5 mg of NPs (with an antibody quantity equal to 41.4 μg/mg of PDLLA) was put in contact with 0.745 mL of a 0.250 mg/mL lipase solution in PBS. Then, 0.255 mL of PBS was added to reach a final volume of 1 mL. The antibody/antigen weight ratio used was 1/3. The suspension was sonicated for 15 min to enable the suspension of NPs and then it was incubated for 1 h under magnetic stirring at 25 °C. Subsequently, centrifugation was carried out under the conditions already described, washing the precipitate twice with distilled water. Finally, the functionalized NPs were recovered, freeze-dried and stored at 4 °C.

### 4. Conclusions

In this study, a conjugation technique via EDC/NHS allowed us to immobilize anti-CRL MoAbs on the surfaces of PDLLA NPs, thus obtaining an innovative enzymatic support consisting of PDLLA NPs conjugated with anti-lipase antibody on which we carried out lipase immobilization tests. CRL non-covalent immobilization by using PDLLA@MoAb NPs produced an activated biocatalyst, as tested in hydrolysis standard assays. Although the use of monoclonal antibodies applied to enzyme immobilization makes the technique expensive, the advantage of using this immobilization method could lie in providing a new approach to the purification of enzymes from crude enzyme preparations and the possibility of reusing the polymeric support. Future studies will be devoted to optimizing lipase immobilization and additionally testing the novel immobilized catalyst in esterasic reactions in organic phase, thus widening the possibility of improving the use of such immobilized enzymes. Moreover, this immobilization strategy, developed for lipases, could be useful also for the stabilization of other classes of enzymes, and it could broaden enzyme applications in other biotechnological fields besides biocatalysis, such as sensors or bioseparations.

**Author Contributions:** Methodology, L.C. and H.R.; investigation, V.C.S.; resources, A.R.; writing—original draft preparation, L.C. and C.P.; writing—review and editing, L.C. and C.P.; supervision, C.P. All authors have read and agreed to the published version of the manuscript.

**Funding:** This research received funding from Sapienza University (Ateneo 2019 grant n° RG11916B427 D180B).

**Data Availability Statement:** The data presented in this study are available on request from the corresponding author. The data are not publicly available due to privacy reasons.

**Conflicts of Interest:** The authors declare no conflict of interest.

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
