# Peer review of "Polymeric Nanoparticles Decorated with Monoclonal Antibodies: A New Immobilization Strategy for Increasing Lipase Activity"

_catalysts, doi:10.3390/catal11060744_

Round 1

Reviewer 1 Report

The unclear points found in the original manuscript have been clarified. This paper is considered acceptable for publishing in its current state.

Author Response

We thank the referee for his appreciation

Reviewer 2 Report

The submitted manuscript entitled “Polymeric nanoparticles decorated with monoclonal antibodies: a new immobilization strategy for increasing lipase activity” presents a method of immobilizing the enzyme on a solid support in order to increase its activity. The manuscript is interesting and has a practical dimension. The use of nanostructured carriers for enzyme immobilization opens the possibility of increasing the surface available for binding with enzymatic proteins and stabilizing the protein itself. The ability to chemically functionalize enzyme carriers appears to be a new approach to specific enzyme binding. The presented studies are a continuation of previous work, and constitute the basis for further study on the stabilization of other classes of enzymes.

Author Response

We thank the reviewer for his/her appreciation

Reviewer 3 Report

In the paper entitled “Polymeric nanoparticles decorated with monoclonal antibodies: a new immobilization strategy for increasing lipase activity”, the authors have reported results related to a new immobilization strategy of lipase with the aim to have a new potential system to be used in biotechnologies.

In this manuscript there are two main problems: the first is in the title, because the authors indicate the enzyme immobilization to increase the lipase activity. Here it is described a new immobilization strategy, partially previously reported by themselves, stressing in the “Introduction” the importance of lipases in biotechnological application and the advantage to use immobilized enzymes to recover it after the use, and to increase the enzymes stability. The question is: do the authors present a new tool of enzyme immobilization for potential biotechnological applications or they are interested to increase the lipase activity? Because in literature are reported many example of increased lipase activity by changing chemical and physical parameters, and in this case authors could be report other example of lipase activity increasing to compare with this new immobilization strategy.

Second, in literature in all the reported case of enzymes immobilization there is an effect of increasing of stability and/or activity. The critical point is the low amount of lipase immobilized, and as reported in the “Conclusion” the system is many expansive and with low yield. Have the authors tried other immobilization systems, including covalent ones, in order to compare the results of enzyme stability and activity, through which it is possible to establish whether the proposed immobilization system is better than the others, even if more expensive?

In conclusion, my opinion is to consider the manuscript after major revisions.

Best Regards

Author Response

A1: In this manuscript there are two main problems: the first is in the title, because the authors indicate the enzyme immobilization to increase the lipase activity. Here it is described a new immobilization strategy, partially previously reported by themselves, stressing in the “Introduction” the importance of lipases in biotechnological application and the advantage to use immobilized enzymes to recover it after the use, and to increase the enzymes stability. The question is: do the authors present a new tool of enzyme immobilization for potential biotechnological applications or they are interested to increase the lipase activity? Because in literature are reported many example of increased lipase activity by changing chemical and physical parameters, and in this case authors could be report other example of lipase activity increasing to compare with this new immobilization strategy.

Q1: We thank the reviewer for giving us the opportunity to better explain the aim of our work. In the present study, we have used our expertise and previous experiences in CRL stabilization through its binding to monoclonal antibodies, as well as its adsorption on nanostructured carriers, with the aim of exploiting our findings to prepare a new immobilization system made of PDLLA NPs functionalized with MoAbs.  The use of nanocarriers provides a high surface area, while the use of MoAbs allows to work with crude commercial enzyme preparations or even with unpurified enzymes. Nonetheless, this new immobilization strategy would allow to broaden enzyme applications in other biotechnological fields besides biocatalysis, such as sensors or bioseparations. We have modified the text accordingly.

A2: Second, in literature in all the reported case of enzymes immobilization there is an effect of increasing of stability and/or activity. The criticalin point is the low amount of lipase immobilized, and as reported in the “Conclusion” the system is many expansive and with low yield. Have the authors tried other immobilization systems, including covalent ones, in order to compare the results of enzyme stability and activity, through which it is possible to establish whether the proposed immobilization system is better than the others, even if more expensive?

Q2: We agree with the referee that in literature there are a lot of examples of lipase immobilization procedures but, as well known, lipase activity and also stability largely depends on the immobilization procedure employed and on the physico-chemical properties of the carriers used. In fact, adsorption via hydrophobic interactions is generally more effective for lipases than for other proteins [Gitlesen et al., Biochim Et Biophys Acta-Lipids Lipid Metab 1997, 1345(2):188–196]. The success of lipase adsorption on a hydrophobic support depends on the structural domain called

lid [Cesarina et al., BMC Biotechnol. 2014, 14, 1472–6750; Xue et al., J. Chin. Chem. Soc. 2018, 66, 427–433]. The lid structure of lipases contributes to provide an open and closed configuration to the enzyme [Ghamgui et al., Biochem. Eng. J. 2007,37, 34–41; Fernandez-Lorente et al., Methods Mol. Biol. 2020, 143–158.]. As well known, in the closed conformation, hydrophobic residues face towards the catalytic site (inside the lid), thus preventing substrate binding. The open conformation of the lid occurs in the presence of a hydrophobic interface. The catalytic site is exposed and binds to the substrate, which causes the phenomenon known as “interfacial activation” [Hirata et al., ChemistrySelect 2016,1, 3259–3270]. The interfacial activation of lipases occurs when the enzyme is adsorbed on the hydrophobic surface and causes a shift of the conformational equilibrium towards the open conformation of the lipase [Bassi et al., Int. J. Biol. Macromol. 2016, 92, 900–909].

The use of MoAbs in the immobilization carrier we have developed allows to bind lipases with a high selectivity and specificity, even if they are present at low concentrations in the reaction medium.  We demonstrated that despite the small amount of bound lipase, it is possible to activate specifically the binded lipase molecules by using strategies able to induce (as in this case study) protein activation by shifting the conformational equilibrium towards the active form of the enzyme. However, the conformational stabilization of an enzyme induced by its interaction with a MoAb could be useful also for other classes of enzymes, since it most probably would result in a higher stability towards denaturing agents. We think that in our case the remarkable increase in lipase activity of CRL immobilized on PDLLA@MoAb NPs could be ascribed to multiple activation phenomena not only related  to the presence of MoAb molecules. In fact, on the basis of the reported data, such activation cannot be merely seen as an addition of the activation phenomena due to the presence of MoAb and PDLLA NPs. These results could be clarified in the future by exploring the number of active sites involved in enzyme-substrate formation.

Reagarding the use of other immobilization procedures we didn’t perform measurements of lipase activity after covalent immobilization on polymeric supports because generally covalent attachment on solid surfaces reduces enzyme stability and requires additional synthetic steps. We have focused our studies on micro or nanocarriers of different dimensions and chemical structure (polymer or metallic based materials) to physically adsorb lipase molecules on their surfaces by using simple and low cost procedures [Chronopoulou et al., International journal of biological macromolecules 2020, 146, 790-797; Venditti et al., Colloids and surfaces B: Biointerfaces 2015, 131, 93-101; Chronopoulou et al., Soft Matter 2011, 7, 2653–2662].

We have modified the text to include some of these considerations.

Round 2

Reviewer 3 Report

the authors have improved the manuscript. Now it is sufficient for publication.

Best regards